# Reproducibility study of "Competition of Mechanisms: Tracing How Language Models Handle Facts and Counterfactuals"

**Tijs Wiegman**                                              *tijs.wiegman@student.uva.nl*
*University of Amsterdam*

**Leyla Perotti**                                             *leyla.perotti@student.uva.nl*
*University of Amsterdam*

**Viktória Pravdová**                                         *viktoria.pravdova@student.uva.nl*
*University of Amsterdam*

**Ori Brand**                                                 *ori.brand@student.uva.nl*
*University of Amsterdam*

**Maria Heuss**                                               *m.c.heuss@uva.nl*
*University of Amsterdam*

**Reviewed on OpenReview:** *https://openreview.net/forum?id=VCG6j3tcAA*

## Abstract

This paper presents a reproducibility study of Ortu et al. (2024), investigating the competition of the factual recall and counterfactual in-context adaptation mechanisms in GPT-2. We extend experiments developed by the original authors with softmax-normalized logits as another metric for gauging the evolution of the scoring of tokens in the model. Our reproduced and extended experiments validate the original paper's main claims regarding the location of the competition of mechanisms in GPT-2, i.e. that the competition emerges predominantly in later layers, and is driven by the attention blocks corresponding to a subset of specialized attention heads. Additionally, we explore intervention strategies based on attention modification to increase factual accuracy. We find that boosting multiple attention heads involved in factual recall simultaneously can have a synergistic effect on factual accuracy, which is further enhanced by the suppression of copy heads. Finally, we rework how the competition of mechanisms is conceptualized and find that the specialized factual recall heads identified by Ortu et al. (2024) act as copy regulators, penalizing counterfactual in-context adaptation and rewarding the copying of factual information.

## 1 Introduction

In recent years, transformer-based language models have shown impressive performance on a variety of natural language processing (NLP) tasks (Min et al., 2023). While great advances in the performance of these models have been made, *understanding* how they operate internally remains a challenge. Mechanistic interpretability (Bereska and Gavves 2024), a field of study which aims to explain the mechanisms in models by reverse engineering algorithms, provides some clues about the existence and nature of these mechanisms. One way in which this can be done is by identifying specific architectural components, also called mechanisms, within the model, such as attention heads or MLP layers, that play crucial roles in generating particular types of predictions (Elhage et al., 2021).

Within the framework of mechanistic interpretability, previous work has laid out the foundation for considering transformer-based language models as a collection of mechanisms that jointly and systematically act on inputs (Olah et al. 2020, Olsson et al. 2022, *inter alia*). Other work has focused on identifying the exact mechanisms found in LLMs as well as their provenance and robustness across different models (Wang et al. 2023, Conmy et al. 2023, *inter alia*). This has enabled the discovery and characterization of single mechanisms such as the copy mechanism (Elhage et al., 2021) and the factual recall mechanism (Yu et al., 2023), which are both explored in this paper.

In this work, we focus on a selection of claims made in Ortu et al. (2024) that are related to the competition of mechanisms in GPT-2. The competition of mechanisms refers to the interaction between the *factual knowledge recall mechanism* and the in-context adaptation or *copy mechanism*. The factual recall mechanism allows models to access and utilize knowledge stored in their parameters during training, enabling them to make predictions based on previously learned facts. Meanwhile, the in-context adaptation or copy mechanism facilitates dynamic updating of model behavior based on immediate context. Essentially, the copy mechanism allows the model to directly use new information introduced via prompting by "copying", even if the new information contradicts the model's stored knowledge. These mechanisms naturally compete when presented with contradictory information, creating a *knowledge conflict* (Xu et al. 2024) in which the model must determine whether to rely on its stored knowledge or adapt to new contextual information.

We attempt to reproduce the results presented by Ortu et al. (2024) and consider the original paper's claims with an alternative metric, softmax-normalized logits. Furthermore, we examine additional strategies for increasing a model's factual accuracy, including separate and simultaneous boosting and suppression of mechanisms involved in factual recall and in-context adaptation. In doing this, we seek to shed some light on two main questions: can we use probabilities instead of logits to understand how the competition of mechanisms evolves in the model and how can we best characterize the attention heads involved in the factual recall and copy mechanisms.

Our contributions can be summarized as follows. First, we find that softmax-normalization does not give significantly different results when probing the importance of different model components obtained using logits, providing evidence for the robustness of logit inspection. Second, we observe synergistic interactions between copy heads, as the effect of suppressing multiple copy heads at once is greater than the sum of suppressing the heads individually; to the best of our knowledge, this has not been discussed in this way in the literature before. Third, we describe a regulational functionality of the copy mechanism mediated by attention heads that supports the copying of factual tokens and penalization of in-context adaptation to incorrect information as an alternative to a separate factual recall mechanism. Finally, we find evidence that the competition of mechanisms proposed by Ortu et al. (2024) is easily perturbed by minor changes in the structure of the data provided to the model.

In this paper, we successfully reproduce the main experiments in Ortu et al. (2024). The scope of our reproduction is outlined in Section 2. We explain the methods adopted in the original paper and in this paper in Section 3, and present our results for the reproduced experiments and further investigation in Section 4. Finally, we discuss our results, reproduction experience, and conclude in Section 5. Our final codebase for our experiments has been made public. [1]

## 2 Scope of reproducibility

We measure the competition between the two mechanisms based on the activations from each of the model's components. We say that a component is highly activated if the magnitude of the difference between the activations of the component on a factual token and a counterfactual token is large.

The claims we discuss in this investigation are:

**Claim 1:** Both the individual mechanisms and competition take place in late, but not early layers.

**Claim 2:** The attention blocks play a larger role in the competition of mechanisms than the MLP blocks.

---

[1]Our code: `https://github.com/LCPerotti/Competition-Mechanism-Reproducibility`

**Claim 3:** A few specialized attention heads contribute the most to the competition.

**Claim 4:** All the highly activated heads attend to the same position: the attribute token.

**Claim 5:** The factual information flows by penalizing the counterfactual attribute rather than promoting the factual subject.

**Claim 6:** Modifying a few selected values in the attention map greatly increases the factual recall, jumping from 4% to 50% for GPT-2.

## 3 Methodology

### 3.1 Problem setup

Following the setup in Ortu et al. (2024), we look into the next token completions of GPT-2 to compare the outcomes of the factual recall and in-context adaptation mechanisms. The competition of mechanisms is elicited by the prompt given to the model. The model contains encoded knowledge in its parametric memory, which can be retrieved by its factual recall mechanism when prompted. The model will also repeat information provided to its context, even if it is not factual. We call this its counterfactual mechanism. By first stating a counterfactual statement, followed by an incomplete repetition of the statement (as is described in Section 3.4), both mechanisms are triggered. The model is then made to decide between two tokens, the token completing the statement factually, $t_{\text{fact}}$, and the token repeating the counterfactual information, $t_{\text{cofa}}$, beginning the competition between the two mechanisms.

We can see which mechanism is preferred by comparing the model's scoring of $t_{\text{fact}}$ vs $t_{\text{cofa}}$ in various locations inside the model. To compare how and where this preference is decided, we analyze how the scores of the target tokens evolve throughout the model's layers and components, utilizing multiple methods for explainability.

### 3.2 Explainability methods

In their paper, Ortu et al. use two methods, logit inspection and attention modification, to track the two competing mechanisms over the model's processing of the input. When reproducing their results, we also utilize these methods.

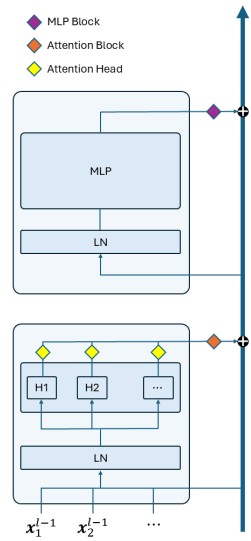

#### 3.2.1 Logit inspection

To trace the competing mechanisms over various positions of the model, the original paper tracks the logit values of the factual and counterfactual tokens, denoted $t_{\text{fact}}$ and $t_{\text{cofa}}$ respectively, using Logit Lens (Nostalgebraist, 2020).

Logit Lens is a method that can give insight into how the model's scoring of tokens changes throughout its layers. Logit Lens works by projecting a model's intermediate representations from the embedding space into the vocabulary space, allowing a view into intermediate logits of vocabulary tokens.

The model's architecture consists of an initial "embedded" representation of the text, followed by an update added by each transformer block. In a transformer model, the final transformer block is followed by two layers: a layer-normalization layer and a linear layer, with the latter represented by the "unembedding matrix". This matrix transforms the hidden size internal embedding into vocabulary-sized logits, representing the model's score for each token in its vocabulary.

Figure 1: The internals of a transformer block, with markers indicating where the attribution Logit Lens interventions take place.

Logit Lens applies these two output layers to intermediate activations positioned in between transformer blocks to calculate intermediate logits.

Logit inspection can be applied at other locations in the model too. The traditional application of Logit Lens is to apply it only after a full transformer block, and this leads to a full set of intermediate logit scores. However, we can also use this to track the changes to the token scores in specific components. We do this by applying the Logit Lens to contribution activations, which are the activations before they are combined with the residual stream, resulting in so-called attribution logits. Figure 1 illustrates the locations where we take the activations to compute attribution logits. Applying Logit Lens after the attention (ATT) and multi-layer perception (MLP) blocks, indicated in Figure 1 in orange and purple respectively, we can follow the influence of these blocks on the token scores. When applied directly after an attention head, indicated in yellow in Figure 1, this shows the specific attention head's attribution to the token scores, allowing to identify precise influences.

A common implementation of Logit Lens is provided by the TransformerLens library (Nanda and Bloom, 2022), which was used in Ortu et al. (2024) and also in our reproduction and further experiments.

### 3.2.2 Attention modification

Attention modification is a type of model intervention executed by modifying the values of attention weights. In particular, when applied to select components, this can be used as a way to understand how information flows through the model by influencing a model's output.

We follow the method outlined by Ortu et al. (2024). That is, we intervene on the attention matrix $A^{hl}$ of the $h$-th head at the $l$-th attention layer. When modifying the attention values, we focus on specific entries in the matrix. Specifically, we focus on the positions $(i, j)$ where $j < i$, which is the attention value of the $i$-th input token attending to the earlier $j$-th input token. The modification is given by Equation 1.

$$A_{ij}^{hl} \leftarrow \alpha A_{ij}^{hl} \tag{1}$$

### 3.3 Model descriptions

Ortu et al. (2024) test their main claims on GPT-2 (Radford et al., 2019). GPT-2 is a decoder-only transformer model with 124 million parameters. It was pre-trained on a large corpus of English data, largely comprised of web pages. GPT-2 is a popular model for mechanistic interpretability as it is smaller in size, accessible, and was one of the earlier transformer-based language models available to the research community. In addition, the mechanisms of GPT-2 are more widely understood as a significant portion of the early work done in mechanistic interpretability was conducted on GPT-2. Ortu et al. (2024) provide supplemental results using Pythia-6.9B (Biderman et al., 2023) to test the generalizability of their claims.

We use the GPT-2 for the reproduction of the results of Ortu et al. (2024) and further experiments. Due to resource constraints, the ability to directly compare our results to Ortu et al. (2024), and the more widely available literature on the inner workings of GPT-2, we focus exclusively on this model.

### 3.4 Datasets

We use two datasets based on COUNTERFACT (Meng et al., 2022). The original COUNTERFACT dataset contains 219,180 tuples of factual relations. These consist of a subject, $s$, and a relation, $r$. For each factual statement $(s, r, \_)$, there are two possible attributes: a correct attribute and an incorrect attribute. Ortu et al. introduce a variant of this dataset, which we call CF-Juxtaposed for notational ease. This dataset was filtered to include only statements where the correct attribute consisted of only a single token, and was predicted as the most likely next token by GPT-2. Again, we denote the correct, factual attribute token as $t_{\text{fact}}$ and the incorrect, counterfactual attribute token by $t_{\text{cofa}}$. A subset of 10,000 randomly chosen samples was produced containing entries where the model would predict the factual token for completion [2].

To prompt the model into activating both its factual and counterfactual mechanism, the full prompt contains the statement with the incorrect attribute $(s, r, t_{\text{cofa}})$, followed by a repetition of the unfinished statement

---

[2]CF-Juxtaposed Dataset: https://huggingface.co/datasets/francescortu/comp-mech

$(s, r, \_)$. The two statements are then connected by prepending "Redefine:" to the prompt. The full prompt template is given in Equation 2.

$$(\text{'Redefine:'}, s, r, t_{\text{cofa}}, s, r, \_) \tag{2}$$

An example given in Ortu et al. (2024) uses the statement "iPhone was developed by _", consisting of the subject $s$ "iPhone", with relation $r$ "was developed by" with correct attribute $t_{\text{fact}}$ "Apple" and incorrect attribute $t_{\text{cofa}}$ "Google". Resulting in the final prompt: *"Redefine: iPhone was developed by Google. iPhone was developed by _"*. We conduct experiments using CF-Juxtaposed for the reproduction and our further experiments from Section 4.6 to Section 4.8.

The second dataset we use is the CounterFact-Tracing (CF-Tracing) dataset developed by Neel Nanda[3]. This version of the dataset contains statements in the form $(s, r, \_)$ with a truthful completion and an incorrect completion obtained from the original COUNTERFACT dataset. The counterpart to the statement above in the style of CF-Tracing is "iPhone was developed by". We highlight that prompts in this dataset do not contain a repetition the statement with the incorrect attribute, and that the competition of mechanisms is driven by incorrect prediction within the model instead, not necessarily adapting to a particular erroneous token as with CF-Juxtaposed. We use CF-Tracing in Section 4.9.

### 3.5 The argument for normalizing logits

Many of the claims in this paper rely on comparing the magnitudes of logits. Specifically, the magnitudes of logits are compared across layers for **Claim 2**, across different layer blocks (ATT and MLP) for **Claim 3**, and across different attention heads for **Claim 4**.

Using intermediate and attribution logits to understand the contributions of the components is a well-founded practice. When a normalization layer is correctly applied, activations are transformed into a zero-mean and unit variance Gaussian distribution for all inputs, meaning that the scale and center of the inputs are controlled. Assuming the normalized activations are projected using a frozen projection layer (unembedding matrix), this forms a systematic, linear transformation with consistent scale on all inputs. As a result, any differences in the logits are solely due to differences in the input data, not fluctuations in the learned model parameters or input scaling. This method is also attractive because linear transformations are easy and cheap to implement (Nanda, 2022).

Nevertheless, investigations of salience methods in neural models have shown that results using logits and probabilities can vary in their faithfulness, lending credence to the possibility that empirical results using probabilities and logits may diverge. The model uses a softmax function to map logits to output probabilities, which are then used in predicting the next token. The model is trained based on these probabilities, not on logits, so any intermediate representation of a token's position in the model is most naturally expressed in probabilities and not as logits.

Magnitudes can significantly change after applying the softmax normalization, depending on the logits of the whole vocabulary. For example, two tokens with similar logits can be squashed even closer together by normalization, or alternatively they can be spread out further. The same is true for tokens with diverging logits, they can be separated further or pulled closer together. The effect of normalization depends on how the logits of the target tokens compare to those of the other tokens. In conclusion, unlike logits, probabilities more closely reflect the *relative* token importance (Bastings et al., 2022), which is what we are trying to measure. For this reason, using both metrics is required for a comprehensive analysis into any claims, as well as it can provide insights into how good each metric is.

## 4 Results & analysis

We begin by reproducing results from Ortu et al. (2024) and analyzing the results. We build on top of the existing experiments with two new lines of exploration. First, we evaluate whether a similar analysis

---

[3]CF-Tracing Dataset: `https://huggingface.co/datasets/NeelNanda/counterfact-tracing`

of the competition of mechanisms can be made when activations are normalized with the softmax function. The second investigative line consists of experiments aimed at better understanding how to characterize the behaviors of attention heads involved in the competition of mechanisms. We use a similar setup as in the original paper for attention modification, but instead we examine how suppressing copy heads alters factual recall. We then analyze the interactions between the copy heads and factual recall heads and determine whether certain properties are intrinsic to factual recall heads. Finally, we look at how models with attention modification behave in more general knowledge recall and model performance tasks.

## Reproduced experiments

To reproduce the results from Ortu et al. (2024), we use the supplementary code[4] provided alongside the paper. After carefully examining the code, we find that the code strongly corresponds to the methods described in the paper. With some tweaks to get the code running, reproducing the original experiments consistently yielded the same results and plots. We follow the same reasoning as the original paper for our analysis of the results, and present our replicated plots in an appendix.

### 4.1 Intermediate logits per layer

To study in which layers of the model the competition of mechanisms takes place, the original paper uses logit inspection to see how the scores for the $t_{\text{cofa}}$ and $t_{\text{fact}}$ change throughout the layers. Specifically, we examine the activations of the model as it predicts the last token position. To do this, intermediate logits are calculated as described in Section 3.2.1 by applying Logit Lens on the activations after every layer (i.e. transformer block). We then take the average logit score over all inputs in the dataset.

Figure 4a in Appendix A shows an upward trend in the average logit scores across layers, as reported in the original paper (Figure 2b of Ortu et al. (2024)). This demonstrates that the relevance of these tokens increases in later but not early layers. We conclude this holds for the individual mechanisms and the competition between, supporting **Claim 1**.

### 4.2 Attribution logits per block

To further analyze which model components are involved in the competition of mechanisms, the original paper investigates the attribution logits of each layer's ATT and MLP blocks. We apply Logit Lens after each block, as described in Section 3.2.1. Again we average the logits over all inputs and focus on the model as it makes predictions for the last token position. We now analyze our results using the difference of the attribution logits of $t_{\text{fact}}$ and $t_{\text{cofa}}$, $\Delta_{\text{cofa}} = \text{Logit}(t_{\text{cofa}}) - \text{Logit}(t_{\text{fact}})$, where the logits are computed per block.

Blocks with larger $|\Delta_{\text{cofa}}|$ are assumed to affect the model's token scores more, implying more involvement in the competition of mechanisms. Just as in Ortu et al. (2024), Figure 4b in Appendix A (Figures 3a and 3b of Ortu et al. (2024)) shows the difference $\Delta_{\text{cofa}}$ is near zero in early layers for both blocks, then increases in favor of $t_{\text{cofa}}$, supporting **Claim 1**. Also, the difference is larger in the ATT blocks, suggesting these blocks have more impact on the competition of mechanisms. We conclude these results also support **Claim 2**.

### 4.3 Attribution logits per attention head

In this experiment, we explore whether particular attention heads are more influential on the competition between $t_{\text{cofa}}$ and $t_{\text{fact}}$. For each layer and every attention head, we inspect the attribution logits using Logit Lens as described in Section 3.2.1, averaging over inputs and focusing on the last token position. We measure the importance of an attention head using $\Delta_{\text{cofa}}$, now taken as the difference of logit attributions within each attention head.

Figure 4c in Appendix A, as in the original paper (Figure 4a of Ortu et al. (2024)), shows only a few specialized heads with higher magnitudes of $\Delta_{\text{cofa}}$, some positive and fewer negative. A large magnitude of $\Delta_{\text{cofa}}$ corresponds to greater involvement in the competition of mechanisms, and since this only occurs on a

---

[4]Code from Ortu et al. (2024): `https://github.com/francescortu/comp-mech/tree/refactor`

few specific heads, we conclude these results support **Claim 3**. As these heads are all located in the later layers, this also supports **Claim 1**.

### 4.4 Attention head inspection

The original paper now more closely examines the attention heads that seem to be of consequence based on the previous experiment. Specifically we focus on L9H6, L9H9, L10H0 and L10H10 which favor $t_{\mathrm{cofa}}$, and L10H7 and L11H10 which favor $t_{\mathrm{fact}}$. We investigate the attention scores between the last token position and the other token positions in the prompts.

Figure 4d in Appendix A shows that the attention scores are concentrated at the attribute position in all the aforementioned heads, in agreement with the original paper (Figure 4b of Ortu et al. (2024)). This supports **Claim 4**. Considering the heads favoring $t_{\mathrm{fact}}$ mostly attend to the attribute position and scarcely to the subject positions, we conclude the factual information penalizes the counterfactual attribute rather than raising the factual subject, confirming **Claim 5**.

### 4.5 Attention modification

We replicated the grid search for $\alpha$ over $\{2, 5, 10, 100\}$ for the two most important factual heads L10H7 and L11H10, as seen in Table 3 in Appendix A, and confirmed $\alpha = 5$ to be the lowest coefficient with a significant impact on factual recall. Beyond $\alpha = 5$, the increase in factual recall is marginal. This supports the results of original paper, specifically **Claim 6**.

## Further experiments

We now introduce new experiments and findings. The primary goals of pursuing these experiments is to investigate whether probabilities provide a similar level of prediction for analyzing the components most heavily involved in the competition of mechanisms, and how we can best describe the behaviors of these components, primarily attention heads, by intervening on the attention values of the models.

### 4.6 Logit normalization with softmax

**Methods**

Using logits may not be optimal and normalizing them could relieve this (see Section 3.5). For this experiment, we extend the experiments in Sections 4.1, 4.2 and 4.3 of this paper by applying softmax normalization (over the whole vocabulary). Mirroring the original paper, we define $\widetilde{\Delta}_{\mathrm{cofa}} = \mathrm{Softmax}\big(\mathrm{Logit}(t_{\mathrm{cofa}})\big) - \mathrm{Softmax}\big(\mathrm{Logit}(t_{\mathrm{fact}})\big)$, the difference between the probabilities of $t_{\mathrm{fact}}$ and $t_{\mathrm{cofa}}$.

**Results**

We find that using the softmax does not change the main trends displayed in logits. The results of each experiment are shown in Figure 2. Figure 2a shows identical trends to Figure 2b in Ortu et al. (2024), as the probabilities grow in the later layers and $t_{\mathrm{cofa}}$ dominates $t_{\mathrm{fact}}$. Note that the data in this figure is shown on a log scale, meaning the divergence between the two curves is much more pronounced. Additionally, the probability of $t_{\mathrm{cofa}}$ gets to about 0.71, meaning its probability is significant and dominates all other tokens. This suggests both mechanisms take place in the late, but not in the early layers, supporting **Claim 1**.

Figure 2b shows the same trends as in Figures 3a and 3b seen in Ortu et al. (2024), as only the later layers seem to contribute and ATT more so than MLP. The magnitude difference between layers is even more significant, as is the magnitude difference between the two block types, ATT and MLP. Thus, this strongly supports **Claim 2**. Curiously, for layers 9 and 10 (very active layers), the difference of $\widetilde{\Delta}_{\mathrm{cofa}}$ between the attention blocks and MLP block is much more prominent than that observed with $\Delta_{\mathrm{cofa}}$. This could be because ATT block is more relevant than MLP in the competition of mechanisms and using softmax shows this more prominently. Alternatively, it could be that the attention block attribution logits have higher variance, which in combination with properties of the softmax function results in bigger $\widetilde{\Delta}_{\mathrm{cofa}}$.

Figure 2c is very similar to what Ortu et al. (2024) presented in Figure 4a. The main difference is not the identity of the heads involved in each mechanism, but rather the strength of their contribution. We can see the heads where $\widetilde{\Delta}_{\text{cofa}}$ is positive mostly have larger magnitude compared to the heads where $\widetilde{\Delta}_{\text{cofa}}$ is negative. The positive head L10H0 is especially divergent from the original plot, as it is very dominant in this plot. In heads where $\widetilde{\Delta}_{\text{cofa}}$ is negative, L11H3 is most dominant in this plot whilst it was least so in the original plot. These differences indicate that the reported relative involvement of each head in the original paper could be limited. However, **Claim 3** is strongly supported by these results, as we see the same set of highly active, specialized heads. Altogether, the experiments using softmax-normalized logits demonstrate that we can draw the same conclusions about the involvement of a model component in the competition of mechanisms using probabilities as those we do with logits.

We continue without using softmax-normalization.

### 4.7 Suppression of the copy mechanism via attention modification

**Methods**

We extend the line of reasoning introduced in Section 4.5 and attempt to increase factual recall through the suppression of the copy mechanism by modifying the attention heads contributing most to this mechanism. In doing so, we hope to learn more about the importance of these attention heads in the competition of mechanisms. We do this in a similar fashion as in Section 4.5, this time selecting the heads responsible for the counterfactual adaptation mechanism. We perform a grid search over the values $\alpha \in \{1, 0.5, 0.2, 0.1, 0\}$ on the four-combination of relevant heads (L7H10, L9H6, L9H9 and L10H0) to determine the best $\alpha$ for further use. Based on heuristic understanding and the results of an exploratory grid search (see Table 5 in Appendix C), we choose a value of $\alpha = 0$ to suppress the relevant attention heads by total ablation, as it should produce the most pronounced suppression effect of any $\alpha \in \mathbb{R}_{\geq 0}$.

We investigate the impact of modifying individual attention heads relevant to factual recall, as well as all subsets of these heads. We separately boost attention heads contributing to the factual recall mechanism and suppress attention heads relevant to in-context adaptation.

**Results**

While the individual boosting of the heads L10H7 and L11H10 each yielded a significant improvement (23.92% and 18.64% respectively), boosting them together yielded even better results at 50.14%, suggesting a powerful synergistic effect when these specific attention heads are modified together.

We identify that the heads that contribute most to counterfactual adaptation are L7H10, L9H9, L9H6 and L10H0, in agreement with the original paper. We investigate the impact of ablation (modification with $\alpha = 0$) of different subsets of these heads on overall factual recall. The results in terms of factual recall % for different subsets can be seen in Table 1. The ablation results reveal an intriguing pattern of head interactions. While individual heads show modest improvements when ablated alone (with L10H0 being the exception at 12.74%), their combined effect is again far greater than the sum of their parts. This is particularly evident with L9H6, and to a much lesser extent with L9H9, which barely improves over baseline when ablated individually (4.19%) or in pairs or triplets, yet contributes significantly to the four-head configuration that achieves 38.78% factual recall. This suggests these heads operate synergistically rather than independently,

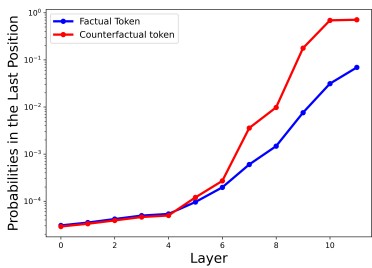

(a) Intermediate probabilities per layer for $t_{\text{fact}}$ and $t_{\text{cofa}}$ in the last position (log scale).

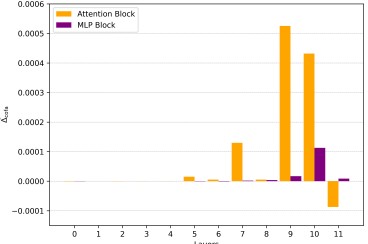

(b) Difference in attribution probabilities $\widetilde{\Delta}_{\text{cofa}}$ for different blocks in the last position.

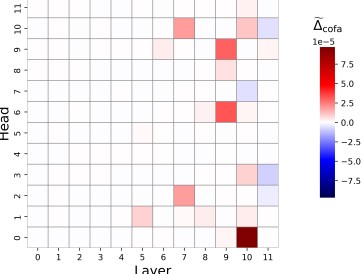

(c) Difference in attribution probabilities $\widetilde{\Delta}_{\text{cofa}}$ for different attention heads in the last positions.

Figure 2: Results of same experiments with logit normalization using softmax.

with their full potential only realized when working in concert. This is especially notable in the performance jump from the best three-head combination (28.19%) to all four heads (38.78%).

### 4.8 Simultaneous factual boosting and copy suppression

**Methods**

We now turn to characterizing attention heads involved in the competition of mechanisms by observing the interaction between them. By doing this, we can further understand how the competition of mechanisms works and how the factual recall mechanism can be boosted. We select the best subsets involved in each mechanism based on the results of Sections 4.5 and 4.7 and evaluate the factuality of responses when boosting and suppressing simultaneously.

**Results**

Through simultaneous modification of multiple attention heads, we discovered that boosting L10H7 and L11H10 while suppressing L7H10, L9H9, L9H6, and L10H0 yields a factual recall rate of 78.81%, as seen in Table 2. This, however, is only 1.2% higher than the 77.63% that we get when suppressing only L7H10, L9H9 and L10H0 while boosting the same heads. Once more, this shows the interplay between attention heads and reinforces our earlier observation about L9H6's unique behavior. While L9H6 plays a crucial role in the complete suppression ensemble, its contribution becomes less significant when combined with boosting interventions, suggesting that the boosted heads may be compensating for its function through alternative pathways by which information is propagated within the model. This observation provides further evidence of the complex interdependencies between attention heads and their ability to adapt and compensate for modifications in the network.

| L7H10 | L9H6 | L9H9 | L10H0 | %fr |
|---|---|---|---|---|
| | | | | **4.13** |
| ↓ | | | | 5.85 |
| | ↓ | | | 4.19 |
| | | ↓ | | 6.13 |
| | | | ↓ | **12.74** |
| ↓ | ↓ | | | 6.21 |
| ↓ | | ↓ | | 10.48 |
| ↓ | | | ↓ | 16.58 |
| | ↓ | ↓ | | 6.28 |
| | ↓ | | ↓ | 18.17 |
| | | ↓ | ↓ | **18.68** |
| ↓ | ↓ | ↓ | | 11.06 |
| ↓ | ↓ | | ↓ | 25.08 |
| ↓ | | ↓ | ↓ | **28.19** |
| | ↓ | ↓ | ↓ | 26.08 |
| ↓ | ↓ | ↓ | ↓ | **38.78** |

Table 1: Factual recall scores for combinations of suppressed heads: L7H10 , L9H6 , L9H9 , L10H0 . (Horizontal lines separate number of heads selected. ↓ denotes suppression).

### 4.9 How many mechanisms?

**Methods**

In the CF-Juxtaposed dataset, we frequently have the case where the factual token occurs as a substring of the subject, that is, sometimes both the factual and counterfactual tokens are present in the prompt given to the model. An example from the dataset is the prompt "Redefine: NBC Nightly News premieres on MTV. NBC Nightly News premieres on", where the factual the factual token, "NBC", appears in the prompt as part of the subject and the counterfactual token, "MTV", appears as the attribute. In fact, over 60% of the prompts in the data from Ortu et al. (2024) are constructed like this. This casts some doubt as to whether the results in Ortu et al. (2024) are the consequence of what they suggest is a factual recall mechanism or the action of the copy mechanism.

We aim to shed more light on which mechanism is responsible for the increased correctness in prediction by examining whether the properties of attention heads we identified as relevant to the factual recall mechanism generalize beyond the structure of CF-Juxtaposed. We perform a similar experiment to Section 4.8 on the CF-Tracing dataset. Recall that CF-Tracing does not have counterfactual statements prepended to its prompts, as described in Section 3.4. We use the attention head combinations that yielded the highest factual recall rates in Section 4.8, that is, we boost L10H7 and L11H10 and suppress L7H10, L9H9, L9H6 and L10H0. Essentially, we ask whether we can still observe the effects of a factual recall mechanism when using a different dataset.

| L7H10 | L9H6 | L9H9 | L10H0 | L11H10 | L10H7 | %fr |
|---|---|---|---|---|---|---|
| | | | | | | **4.13** |
| ↓ | | ↓ | ↓ | | | 28.19 |
| | ↓ | ↓ | ↓ | | | 26.08 |
| ↓ | ↓ | ↓ | ↓ | | | **38.78** |
| | | | | ↑ | | 18.64 |
| | | | | | ↑ | 23.92 |
| | | | | ↑ | ↑ | **50.14** |
| ↓ | | ↓ | ↓ | ↑ | ↑ | 77.63 |
| | ↓ | ↓ | ↓ | ↑ | ↑ | 74.74 |
| ↓ | ↓ | ↓ | ↓ | ↑ | ↑ | **78.81** |

Table 2: Factual recall scores for simultaneously boosting and suppressing attention heads: L7H10 , L9H6 , L9H9 , L10H0 , L11H10 , L10H7 . (↓ and ↑ respectively denote suppression and boosting).

In addition, to control for the repetition of factual token in the subject string, we split the CF-Tracing dataset into two disjoint subsets based on whether the subject contains the factual token. We end up with a subset of of 2006 prompts in which the factual token appears in the subject, and a subset of 19913 samples free of the factual token. We call the aforementioned subsets the *copyable* and *filtered* sets, respectively. If an independent factual recall mechanism exists, we expect to see that the number of correct predictions increases on both subsets of CF-Tracing when intervening on the attention heads listed above.

**Results**

In the copyable set, where the factual token is already present in the prompt as part of the subject, we find that boosting the attention heads involved in factual recall increases the accuracy of the model in predicting the correct token from 21.1% to 24.4%. We don't observe any meaningful change in accuracy in the filtered set when boosting factual heads.

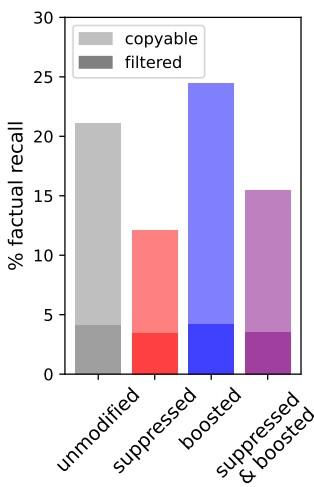

Suppression of attention heads involved in the copy mechanism yielded more interesting results. Even in the presence of the correct token, the suppression of copy heads causes a reduction of the model's accuracy from 21.1% to 12.1%, and to 15.4% when the suppression is combined with factual boosting, as seen in Figure 3. In the filtered set, suppression also reduces factual recall from 4.1% to 3.4%. Simultaneously boosting doesn't correct this behavior and we observe a reduction to 3.5%.

Figure 3: Percentage of factual recall on the CF-Tracing dataset.

Based on the results of this experiment, we believe that the characterization of the attention heads by Ortu et al. (2024) is flawed and occluded by the structure and inherent repetition of the factual token. Instead, we observe that the attention heads Ortu et al. identify as factual recall heads come into action only when copying takes place, as seen for the copyable set in Figure 3. In addition, the effects of intervening on these attention heads has a much smaller effect than in the setting using CF-Juxtaposed. We reason that these attention heads may act as copy regulators, penalizing the copying of counterfactual information, as developed through **Claim 5** by Ortu et al. (2024) and our experiments. Likewise, these attention heads encourage the copying of facts, as demonstrated by the counteracting of copy suppression in Figure 3.

Nevertheless, seeing as these attention heads do not universally correspond to factual correctness, we cast doubt as to whether Ortu et al. (2024) demonstrates that a competition of mechanisms exists in practice. In particular, we argue that the copy mechanism in GPT-2 explains the behaviors observed in Ortu et al. (2024) and in our experiments using CF-Juxtaposed.

### 4.10 Surprisal under attention modification

**Methods**

A natural question that emerges when modifying the attention patterns of a model is whether the behavior of the model on other tasks changes as well. To compare the performance of the model after attention modification, we measure the perplexity of the baseline and attention-modified models on the test split of WikiText-2 with a batch size of 32 and stride of 512. We make the same attention modifications as in Section 4.9.

**Results**

We determine that the performance of the model does not degrade when modifying the attention heads. We obtain a perplexity value of 29.9 over the WikiText-2 test set for both the unmodified and modified models. We reason this result is the way it is because very few entries in the attention head matrices are modified and also due to the specialization of the attention heads in each of the mechanisms. Nevertheless, more work, potentially of a more qualitative nature, is required to better understand how the performance of the model changes when intervening on its attention values.

# 5 Discussion

## 5.1 Further experiments

### Softmax normalization

Applying the softmax normalization in Section 4.6, the experiments of Figures 2a and 2b resulted in near identical trends. This suggests comparing the logits directly is a valid approach for drawing conclusions as was done by Ortu et al. (2024). In Figure 2c, however, there is a difference in which heads are most active, as discussed in Section 4.6.

Figure 4a of Ortu et al. (2024) suggests that L11H10 is the attention head most involved in the factual recall mechanism, whereas the softmax-normalized plot (Figure 2c) indicates that L11H3 contributes somewhat more the competition. In our informal experimentation, L11H10 had a bigger impact on the factuality than L11H3 when boosting the heads involved in factual recall. This indicates that the softmax-normalized logits might not have the same predictive power as logits in determining the degree to which attention heads are involved in the competition mechanism. Nevertheless, this softmax-normalized logits predicted the strength of L10H0 in the adaptation mechanism more accurately than plain logits. More work is needed to better understand the practical advantages and disadvantages of each metric.

There are some interesting comparisons to be made between our results and prior literature regarding the heads of the model. For example, L11H10 and L10H7 have been identified as "negative mover heads", related to copy suppression (Wang et al. 2023, McDougall et al. 2024), which is in agreement with our conclusions. L11H3, however, has been linked to semantic understanding (Lee et al., 2024), which we did not explicitly confirm or reject.

### Attention modification

The compounding effect on factual recall from applying suppression to both L10H7 and L11H10 simultaneously, as mentioned in Section 4.7, hints at synergistic effects. We hypothesize that this is related to the fact that the heads are positioned in different layers, meaning that as the activations propagate, the processing of the first head impacts the inputs of the second head. Thus the processing of the second head is affected by the first head, resulting in behavior separate from the behavior of the heads individually. It is possible that during training, these combined heads are largely responsible for promoting the counterfactual mechanism over the factual one.

## 5.2 Reproducibility experience

Some issues that we faced when reproducing the results from Ortu et al. (2024) were understanding parts of the code. We had to first debug some of the code in order to run their original experiments. In particular, we found the code for experiments using the Logit Lens to be lacking in documentation and challenging to navigate.

In addition, when adapting the logit attribution experiments for using softmax-normalized attributions we ran into issues of scaling up. Instead of calculating the scores for just the target tokens, we now had to calculate the scores for the entire vocabulary to compute the normalized logits, increasing computation and memory demands. We resolved this issue by running the experiment on a computer cluster and by reducing the batch size to 5.

## 5.3 Conclusion

### Outcomes

This study successfully reproduces the experiments of Ortu et al. (2024) and extends the analysis with additional experiments. Our results confirm and strengthen some key claims from the original work, but reveal a more nuanced picture when exploring on new data.

Within the framework introduced by Ortu et al. (2024), we are able to show that the competition of mechanisms can also be analyzed using softmax normalized logits and that the results mostly agree with the analysis done using logits. On the task formulated in this framework, boosting multiple factual heads produces a greater effect on the factual recall suggesting synergistic interactions between factual recall heads. In addition, combining factual boosting and counterfactual suppression significantly increases factual recall, achieving a factual recall of up 78.78%.

Beyond the framework, we find that boosting factual recall heads doesn't improve factual recall in prompts from the COUNTERFACT dataset unless the factual token is present somewhere in the prompt. When the factual token is found in the prompt, factual recall heads align the copying behavior with the model's stored knowledge. These findings characterize the factual recall heads not as promoters of factuality, but more specifically as regulators of factual information within the scope of copying. At the same time, suppressing copy heads significantly reduces factual recall, even in the presence of a factual token. The reduction in factual recall occurs even when the factual recall heads are boosted, indicating that the copy mechanism is considerably stronger than the copy regulation performed by the factual recall heads. We also find that attention modification doesn't cause the model's performance, as measured by perplexity on WikiText-2, to degrade.

**Limitations**

One of the main limitations is the generalizability of our results. Our findings are restricted to one model, GPT-2, and following a similar methodology with another model may yield different results. Another limitation is the dataset we use in Sections 4.7 to 4.8. Over 60% of entries contain the factual token as a part of the prompt, as described in Section 4.9. This makes some of our observations about the competition of mechanisms less precise, as the copy mechanism may serve both counterfactual in-context adaptation and the copying of factual information that aligns with knowledge stored in the model.

Additionally, this work heavily relies on two mechanistic interpretability techniques, namely Logit Lens and attention modification. This can be problematic as these methods may not fully capture model behavior and do not provide a clear causal interpretation of the model's prediction. This investigation's ability to analyze how components interact and how is somewhat narrow, and could have been strengthened by accompanying it with newer methods or different types of explainability techniques. Furthermore, the dataset limits the investigation to predictions comprised of one token on a rather artificial prompting structure.

**Future work**

A potential future investigation could adopt more recent mechanistic interpretability techniques like circuit analysis (Olah et al. 2020, Elhage et al. 2021). In light of Section 4.9 in particular, it would also be interesting to see if the competition of mechanisms could be reimagined in terms of circuits and whether the roles of the factual recall and copy heads we identify make it so that these heads are a part of a single circuit or multiple competing circuits.

Other mechanistic interpretability methods can be applied to investigate the competition of mechanisms. For example, the Logit Lens technique is being further developed and newer, more advanced tools exist, like Tuned Lens (Belrose et al., 2023) and Logit Prisms (Nguyen, 2024).

Our experiments are based on the task of single token prediction as a clearcut way of examining the competition of mechanisms, however, this competition is a feature of other types of tasks as well. Further work to investigate how our analysis of the competition of mechanisms generalizes to other tasks such as multi-token prediction or text summarization could be an interesting extension, depending on the capabilities of the model considered.

Additional future work follows from what was proposed in the original paper. The competition of mechanisms could be examined on larger models. The original paper performed additional tests on Pythia-6.9B, but we were not able to reproduce these due to resource constraints. Another possible route of future work could consider the design of the prompt beyond its current structure. Finally, more investigation is needed to understand how the synergistic interaction works, where it originates, what its exact purpose and behavior is, and if more such interactions exist in the model.

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

# A    Reproduced figures and tables

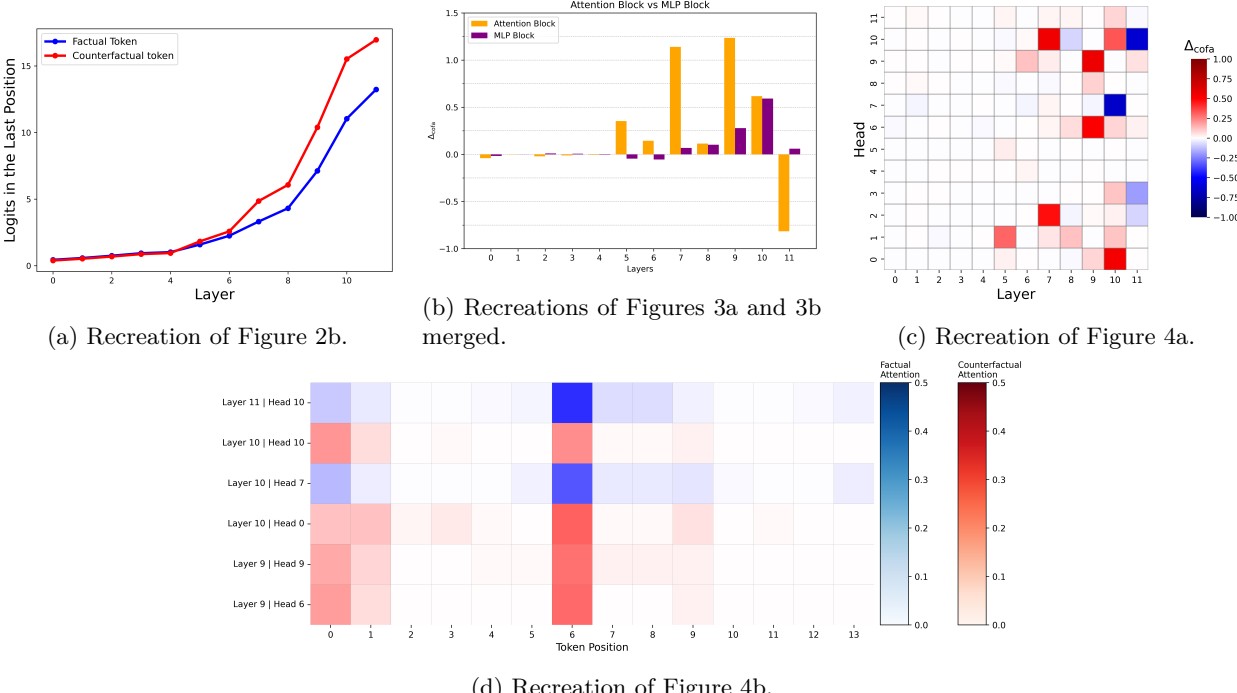

(a) Recreation of Figure 2b.

(b) Recreations of Figures 3a and 3b merged.

(c) Recreation of Figure 4a.

(d) Recreation of Figure 4b.

Figure 4: Recreated figures from Ortu et al. (2024).

| $\alpha$ | factual recall% |
|---|---|
| 0 | 0.67 |
| 1 | 4.13 |
| 2 | 32.18 |
| 5 | 50.14 |
| 10 | 51.56 |
| 100 | 52.29 |

Table 3: Percentage of factual recall with boosting factor $\alpha$.

# B    Baselines for factual boosting

**Methods**

To establish baselines for the efficacy of attention modification as a strategy for increasing factual recall, we examine how boosting individual attention heads across the model affects the percentage of factual responses produced by the model. In performing this experiment, we seek to answer if we can increase factual recall by boosting random attention heads or if this capability is intrinsic exclusively to the identified factual recall heads. We chose 100 random attention heads (seed=347) from various layers in the model, excluding the attention heads we identified as highly involved in the competition of mechanisms. In addition, we boost random pairs of attention heads located in different locations within the model, following Section 4.5 and using a boosting factor of $\alpha = 5$. For both individual heads (Figure 5) and pairs of attention heads (Table 4), we examine interventions of the attention patterns in the attribute position and all token positions attended by the last token position.

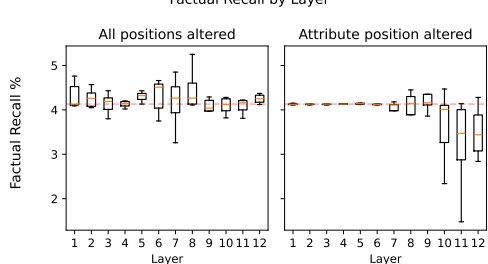

Figure 5:   Box plot of factual recall after modifying randomly selected attention heads per layer, with 5 to 10 heads altered per layer.

**Results**

Figure 5 shows that random boosting leads to factual accuracy between 1% and 6%. We see less variability in factual accuracy when modifying all token positions across all model layers compared with modifying only the attribute position. In addition, when only modifying the attribute we see that the range of accuracies in later layers is larger than in the earlier layers. This is another indication that the competition occurs in the later layers of the model and not in the early layers, supporting **Claim 1**.

| Distance between head pairs | All (Mean %fr) | Attribute (Mean %fr) |
|---|---|---|
| Same Layer ($d = 0$) | 4.49 | 6.68 |
| Consecutive ($d = 1$) | 4.02 | 3.53 |
| $1 < d < 5$ | 4.53 | 4.07 |
| $d > 5$ | 4.24 | 5.42 |

Table 4:   Random boosting of attention head pairs (pairwise comparisons).

We also note that boosting random head pairs does not increase factual recall, regardless of the distance between the heads in a pair and whether the modification was applied to all positions or only the attribute position. When boosting head pairs at the the attribute position, we find that the mean factual recall ranges between 4% and 4.6%. Random modifications of head pairs at all token positions result in mean factual accuracies between 3.5% and 7%, depending on the distance between the heads as seen in Table 4. Once more, we observe greater variability in factual accuracy when modifying all token positions. Additionally, while boosting at the attribute position we observe higher mean factual recall for some distance values, there is no clear trend based on distance.

Importantly, both trials demonstrate that random boosting of single heads or head pairs does not cause the factual accuracy to deviate significantly from that of the unmodified model. This implies that the attention heads observed in Section 4.4 are indeed specialized to the factual recall process and further validates **Claim 3**.

## C   Supplemental experiments

| $\alpha$ | factual recall% |
|---|---|
| 1 | 4.13 |
| 0.5 | 21.38 |
| 0.2 | 34.22 |
| 0.1 | 36.7 |
| 0 | 38.78 |

Table 5: Percentage of factual recall with suppressing factor $\alpha$.

