# OpenReview forum: "Reproducibility study of "Competition of Mechanisms: Tracing How Language Models Handle Facts and Counterfactuals""
_TMLR — Accepted by TMLR_

### Review · Reviewer_vYuB · 2025-03-30

**Summary Of Contributions:**

The paper tries to replicate the findings of Ortu et al. (2024), concerning the so-called “competition of mechanisms” in GPT-2, with a particular focus on **factual recall** vs **counterfactual adaptation**.

Hence, the authors reproduce the original experiments and throw in a few extra plots using softmax-normalised logits. There is also some minor tinkering with attention heads and an overuse of vocabulary about copying and recall.

**Audience:**

Yes

**Broader Impact Concerns:**

None really. It’s hard to see what actual impact this paper would have, given that it largely echoes prior work with little added value. However, note that I may have made a mistake and not really see the improvemtns so I kindly ask you for clarification.

**Claims And Evidence:**

No

**Requested Changes:**

The authors could try to write more clearly, avoid nonessential duplication of the original study, and maybe try a more innovative or critical perspective.

**Strengths And Weaknesses:**

**Strengths:**

- The code seems to run.

- Figures look somewhat similar to those in the original paper.

**Weaknesses:**

- The paper is bloated with excessively technical detail that obscures rather than clarifies the goals.

- There is far too much focus on confirming what was already shown instead of offering new insights.

- The added experiments with softmax-normalised logits feel arbitrary and do not meaningfully advance the discussion (this should be the crucial improvements).

- Much of the analysis is repetitive and offers no novel conclusions.

To summarize: The paper aims to replicate Ortu et al. (2024), examining some behaviour in GPT-2. The reproduction is technically sound and includes some additional analysis, the work lacks originality and offers little beyond confirming existing findings. The heavy technical detail detracts from clarity, and the analysis feels repetitive. Overall, the work is derivative, lacks originality, and does not sufficiently motivate why this replication is worth reading.

---

### Review · Reviewer_u2Nt · 2025-04-04

**Summary Of Contributions:**

1. The authors successfully reproduced findings from Ortu et al. (2024)’s

2. They discuss concern of using logits or softmax probabilities as an alternative metric to better reflect token salience during competition. The conclusion is that using normalized probabilities leads to the same conclusion as the original paper.

3. Additionally, they experiment with intervention strategies to boost factual accuracy by boosting attention heads involved in factual recall. In contrast, random attention modification does not significantly affect factual recall, confirming that the competition is driven by the attention blocks corresponding to a subset of specialized attention heads.

**Audience:**

Yes

**Broader Impact Concerns:**

No concerns on the ethical implications of the work.

**Claims And Evidence:**

No

**Requested Changes:**

1. Related to weakness 1, I think the reproducibility study would be interesting if the work can show the generalizability of published method (different models or datasets), or a broader set of analysis on whether using logits or softmax matter in different MI methods.

**Strengths And Weaknesses:**

1. Following the guideline for TMLR ("A proper reproducibility report that systematically studies the robustness or generalizability of a published method and lays out actionable lessons for its audience could satisfy this criterion."), I think a reproducibility paper should discuss whether the published work is generalizable and what are actionable lessons. In this work, the authors mostly downloaded the code, and inspected their implementation and reproduced their results. Authors use the same model and dataset. It seems to me that it is a rather trivial reproducibility paper. While it is interesting to examine whether normalizing logits to probabilities it a valid concern, it would be interesting to discuss for other papers and methods, whether using logits versus softmax probabilities would or would not change experiment outcomes and conclusions.

2. In the original paper, authors experiment with different alpha values for supression. However, authors then use alpha=0 and claim "we perform suppression by ablation as it should produce the most pronounced suppression effect of any alpha > 0". Even though the intuition makes sense, the methodology is not thorough.

---

> ### Comment · Reviewer_u2Nt · 2025-05-02
> **Reponse**
>
> Thank you for the response. I have carefully read through other reviewer's comments and I think the two valuable contributions are
>
> (1) the discussion regarding logits normalization on experiment outcomes
>
> (2) Section 4.9 pointing out an experimental flaw in the original paper Ortu et al. 2024.
>
> Regarding (1), I think the paper would be interesting to people if the results have broader implication that is not restricted to one particular paper. Regarding (2), I think people who are interested in studying how LLM represent facts and how it interacts with copying heads will take a second thought on the dataset proposed by Ortu et al.
>
> I have updated my recommendations that audience is yes. I still think the paper would be a stronger reproducibility paper if it can generalize results to different models at least.

---

### Review · Reviewer_CthS · 2025-04-10

**Summary Of Contributions:**

The authors present a thorough reproducibility study of the paper "Competition of Mechanisms: Tracing How Language Models Handle Facts and Counterfactuals" by Ortu et al. They successfully replicate most of the original findings and extend the analysis with experiments that employ alternative methodologies. Notably, they demonstrate that the core claims of Ortu et al. remain valid when comparing the model’s preference for factual versus counterfactual tokens using softmax probabilities instead of logits. Additionally, the authors contribute novel insights into the attention modification approach proposed by Ortu et al. by investigating the impact of suppressing attention heads associated with the copy (counterfactual) mechanism. They further analyze the effects of suppressing or boosting the copy and recall heads both independently and in combination. A particularly interesting finding concerns a potential issue in the original dataset: the subject string often contains the factual token, which may have influenced the observed outcomes. By isolating a subset of the data where the factual token is not present in the subject, the authors show that the attention modification technique no longer improves factual accuracy, suggesting that the original results may, in part, be driven by this artifact.

**Audience:**

Yes

**Broader Impact Concerns:**

I don't think there are any major ethical implications of the work that would require adding a broader impact statement.

**Claims And Evidence:**

Yes

**Requested Changes:**

Please refer to the weakness section above.

**Strengths And Weaknesses:**

**Strengths**

- The paper presents several novel and valuable contributions beyond the original work of Ortu et al. The analysis using softmax probabilities instead of LogitLens is particularly compelling, as it strengthens the credibility of Ortu et al.’s claims through an alternative and arguably more interpretable lens. Moreover, the experiments involving the suppression of attention heads most associated with the copy mechanism, along with well-designed baselines for boosting heads relevant to the factual mechanism, significantly enhance the robustness of the original claims through careful ablation studies.

- Section 4.10 offers an especially important and insightful finding: nearly 60% of the prompts in the dataset used by Ortu et al. include the factual token as part of the subject itself. This raises critical concerns about potential confounding effects, as the performance of the attention modification technique differs substantially between the filtered and copyable subsets. This observation has major implications for interpreting the results of the original study.

- The paper is well-written and provides a clear, detailed overview of both the problem and the methodology proposed by Ortu et al. The experimental section is thoughtfully organized and makes direct comparisons to the original results, which facilitates a transparent and coherent evaluation of the reproduced findings. The novel methodologies and experiments are well-motivated, and the authors offer meaningful discussions on the implications of their empirical observations.


**Weaknesses**

- A minor criticism is that some findings could benefit from deeper explanation. For instance, in Section 4.6, the authors state that "the softmax-normalized result suggests that attention blocks are even more important in the competition mechanism than what was presented by the original paper." However, the rationale behind this observation is unclear. Why would using softmax probabilities instead of logits emphasize the role of attention blocks more strongly? A more detailed discussion of this phenomenon would greatly enhance the reader’s understanding.

- The description of the experimental setup in Section 4.10 is somewhat confusing. The authors write, "we perform the prediction of tokens on a variant of the CounterFact dataset, which does not have counterfactual statements prepended to its prompts." If taken literally, this would imply that the model only sees prompts such as "iPhone was created by _", without the counterfactual context. In such cases, it’s unclear why or how a competition of mechanisms would arise at all. This may be a typo, and perhaps the authors intended to say that the factual token is not present in the subject of the counterfactual statement, while the counterfactual prompt remains prepended. Clarifying this would help avoid misinterpretation.

- In the original paper, Ortu et al. focus on instances where the counterfactual token is selected (which reportedly occurs in 96% of cases). It would be helpful to know whether the authors of the current study adopt the same strategy. Additionally, if they did not analyze cases where the factual token is chosen, a discussion explaining this decision would be valuable. Would the original claims—such as the competition between mechanisms occurring earlier in the network and being more prominent in attention layers—still hold in those scenarios?

- For the experiments in Sections 4.7 to 4.10, it is not clear whether the authors are using the logit-based inspection method or the softmax-normalized approach. This detail is important and currently easy to miss. It would be helpful to clearly state the methodology used in these sections.

- Finally, it would be a useful addition if the authors could summarize the tweaks or adjustments they made to get the original codebase from Ortu et al. running. Even a brief mention or a link to a GitHub file documenting the changes would be greatly appreciated by researchers attempting to build on this work.

---

> ### Comment · Reviewer_CthS · 2025-05-06
> **Reply to Rebuttal**
>
> Thanks for the detailed rebuttal! My concerns have been addressed, and I believe the paper does a good job at reproducing the original work by Ortu et al. and provides significant novel findings as well!

---

### Author Response · Authors · 2025-04-24
**Reply (Part 1)**

**Introduction**
We thank the reviewers for their valuable time and feedback. We have carefully examined your criticism and processed it to make revisions to our paper and craft this response to address some unclear points, concerns, and explain how we incorporated the requested changes.

First, we will describe the context and goals of this paper. Also, we want to address the novelty concerns and restate the contributions of this work and emphasise the value. Lastly, we want to illuminate and explain some aspects of the paper and discuss the revisions made to enhance clarity.

**Paper Context** Our goal for this paper is to participate in the MLRC conference for reproducibility and replicability (in collaboration with TMLR). As such, the paper starts by reproducing and verifying the results and analysis of the original paper by Ortu et al. We follow this with a replicability experiment, by testing the original claims using a new metric, the softmax-normalised logits, extending the original methods. We also dig deeper into the setup used by Ortu et al., uncovering significant flaws in the dataset and challenging the notion of a competition of mechanisms framework. Last, we use attention modification tools to further probe and enhance understanding of the mechanisms. Not only do we find actionable ways to use intervention, showing that boosting and suppressing certain heads can increase factual recall, but we also deliver more insight about how the mechanism works, with our results uncovering synergistic effects.

Our paper is mostly focused on robustness, as a test of Ortu et al.'s claims under a more comprehensive evaluation framework. By closely mirroring the original setting (using a similar model, data, and prediction tasks) we isolate methodological effects and assess the stability of the original paper's results. We believe that this focused reproduction and replication allows us to draw strong and insightful conclusions, and leave generalizability for subsequent research.

**Original Contributions**
 We will now summarise what we believe are the key findings of our paper that are valuable contributions to the audience of TMLR and the community at large.
- We find that softmax-normalization does not give significantly different results when probing the importance of different model components obtained using logits, providing evidence for the robustness of logit inspection.
- We observe synergistic interactions between copy heads, as the effect of suppressing multiple copy heads at once is greater than the sum of suppressing the heads individually. To the best of our knowledge, this has not been discussed in this way in the literature before.
- We describe a regulational functionality of the copy mechanism mediated by attention heads that supports the copying of factual tokens and penalization of in-context adaptation to incorrect information.
- We find evidence that the competition of mechanisms proposed by Ortu et al. is easily perturbed by minor changes in the structure of the data provided to the model. While we can't tell for sure that a factual recall mechanism doesn't exist altogether, we find compelling evidence to claim that it doesn't work as suggested by Ortu et al.

---

### Author Response · Authors · 2025-04-24
**Reply (Part 2)**

**Clarifications**

**vYuB** *The added experiments with softmax-normalised logits feel arbitrary and do not meaningfully advance the discussion (this should be the crucial improvements).*

The goal of the softmax-normalized experiments is to understand the limits of the logit inspection method for measuring the importance of model components when making predictions. We chose this normalization method since we think softmax-normalized logits make an appropriate choice for this measure of importance, in particular as models are trained using probabilities and not logits for the next-token prediction task, as we discuss in Section 3.5. Findings in the literature have shown that results using different interpretability methods can vary depending on whether softmax-normalization is used (e.g. Bastings et al. 2022). We adopt this paradigm to test whether logit inspection is a robust method for understanding the participation of different model components. We find that there is no major difference on the results, except that the importance of  attention heads over MLP blocks is more clearly observed using softmax-normalization (see edits to Section 4.6), adding some credence to the robustness of logit inspection.

**u2Nt** *In the original paper, authors experiment with different alpha values for suppression. However, authors then use alpha=0 and claim "we perform suppression by ablation as it should produce the most pronounced suppression effect of any alpha > 0". Even though the intuition makes sense, the methodology is not thorough.*

We added a grid search over suppression values for suppression of all heads together conducted in an early iteration of the paper as an appendix and served as a sanity check. We see this result as somewhat trivial when considering the ample literature on the role of copy heads in GPT-2, especially compared to less explored factual recall heads for which the grid search was done in the original paper.

**CthS** *The description of the experimental setup in Section 4.9 [4.10] is somewhat confusing. The authors write, "we perform the prediction of tokens on a variant of the CounterFact dataset, which does not have counterfactual statements prepended to its prompts." If taken literally, this would imply that the model only sees prompts such as "iPhone was created by \_", without the counterfactual context. In such cases, it’s unclear why or how a competition of mechanisms would arise at all. This may be a typo, and perhaps the authors intended to say that the factual token is not present in the subject of the counterfactual statement, while the counterfactual prompt remains prepended. Clarifying this would help avoid misinterpretation.*

The literal interpretation of this section is indeed the correct one. We have revised this section and Section 3.4 to describe aims of the experiment and the data we use in greater detail. In Section 4.9 does not focus on the competition of mechanisms, but instead on the central claim made by Ortu et al. that there is an independent factual recall mechanism. We hypothesize that the data they use might cause a confusion as to what mechanism plays a role in the results. We design the experiment in Section 4.9 to isolate the purported factual recall mechanism but do not observe the expected effects in this setting, challenging the existence of such a mechanism as described by Ortu et al.

**CthS** *In the original paper, Ortu et al. focus on instances where the counterfactual token is selected (which reportedly occurs in 96\% of cases). It would be helpful to know whether the authors of the current study adopt the same strategy. [...]*

We clarify that the dataset constructed by Ortu et al. (which we now call CF-Juxtaposed) is made from examples in which GPT-2 was able to predict the correct token *without* the counterfactual statement prepended to it (i.e. in the style of CF-Tracing). The figure of 96\% refers to the percent of data samples in which GPT-2 favors the counterfactual token over the factual token *after* counterfactual statements were added to the data. We adopt the same strategy and use the same data up to Section 4.9. The data in CF-Tracing does not have this guarantee.

**CthS** *For the experiments in Sections 4.7 to 4.9 [4.10], it is not clear whether the authors are using the logit-based inspection method or the softmax-normalized approach. This detail is important and currently easy to miss. It would be helpful to clearly state the methodology used in these sections.*

Only minimal logit inspection is being done in these sections, as in these, in line with the original paper, we analyse factual recall over the entire dataset, which solely represents which token was scored higher in terms of the raw logit scores. Even so, since softmax is monotonic, this would not affect the outcome, and as such we do not think this needs to be added.

---

### Decision · Action_Editor_kKks · 2025-06-10

**Recommendation:** Accept with minor revision

**Additional Comments:**

For the final version, it would be helpful if the authors can add the summary of original contributions given in the [response](https://openreview.net/forum?id=VCG6j3tcAA&noteId=RQ0ps3T92M), either at the end of Section 1 or in Section 2.

**Audience:**

Yes

**Audience Explanation:**

People working in mechanistic interpretability may find the detailed explanation of the study useful, and the result of this study provide more robustness to the findings of Ortu et al. Additionally, there are novel findings/contributions that may be useful for the community, such as the use of different metrics, or issues with datasets.

**Claims And Evidence:**

Yes

**Claims Explanation:**

The reproducibility study is accurate and the methodology is clear.